# Improving the Wear Resistance Properties of 7A04 Aluminum Alloy with Three Surface Modification Coatings

Jinmeng Hu [1], Cheng Zhang [1,2], Xiaodong Wang [1], Xiaobo Meng [1], Caihong Dou [1], Hua Yu [1,2], Changji Wang [1,2], Jun Xue [3], Ziping Qiao [3,*] and Tao Jiang [1,4,*]

[1] National Joint Engineering Research Center for Abrasion Control and Molding of Metal Materials, School of Materials Science and Engineering, Henan University of Science and Technology, Luoyang 471000, China; 15638265067@163.com (J.H.); zhangch06@126.com (C.Z.); nmxdwang@163.com (X.W.); mengxiao6o@163.com (X.M.); dch1805@163.com (C.D.); kedayuhua@126.com (H.Y.); wchj_1989@haust.edu.cn (C.W.)
[2] Longmen Laboratory, Luoyang 471003, China
[3] Science and Technology on Transit Impact Laboratory, No. 208 Research Institute of China Ordnance Industries, Beijing 102202, China; oil1999@163.com
[4] Luoyang Wanji Aluminum Processing Co., Ltd., Changjiang Avenue, Industrial Agglomeration Zone, Luoyang 471800, China
* Correspondence: maryqiao@163.com (Z.Q.); tedivy@163.com (T.J.)

**Abstract:** Multiple advantages, such as good formability, high specific strength, excellent thermal conductivity, and high corrosion resistance, enable aluminum alloy wide application in various fields; however, low surface hardness and poor wear resistance limit its further development. In this study, three surface modification coatings were successfully prepared on the surface of 7A04 aluminum alloy by microarc oxidation (MAO) and a combination of hard anodizing treatment (HA) and physical vapor deposition (PVD), named MAO, HA+W+DLC, and HA+Ti+ta-C, respectively. The microstructure, hardness, and tribological properties of the three coatings and the 7A04 aluminum alloy substrate were studied. The results show that the surface quality and hardness of the coated samples were higher than those of the 7A04 aluminum alloy and that the HA+Ti+ta-C coating possessed the highest hardness of 34.23 GPa. Moreover, the wear resistance of the two multilayer coatings was significantly improved during the ring-block wear tests under oil lubrication, exhibiting a wear rate of $1.51 \times 10^{-7}$ mm$^3$/N·m for HA+W+DLC and $1.36 \times 10^{-7}$ mm$^3$/N·m for HA+Ti+ta-C.

**Keywords:** aluminum alloy; MAO coating; HA+W+DLC multilayer coating; HA+Ti+ta-C multilayer coating; wear resistance

## 1. Introduction

Aluminum alloys are widely used in the aerospace industry, especially in commercial transport aircraft and military fighter aircraft, owing to their excellent properties, such as good formability, light weight, high specific strength, and low density (approximately 2.7 g/cm$^3$) [1–5]. However, the low surface hardness and poor wear resistance of aluminum alloys severely limit their application. 7A04 is a kind of superhard aluminum alloy, the dosage of which is second only to steel in modern industry, and surface modification can significantly improve its performance. In recent years, the surface modification of 7A04 aluminum alloy has become an active research subject owing to its potential to improve the wear resistance of aerospace aluminum alloy parts to ensure a long service life, offer good reliability and production quality, reduce production costs, and obtain good economic benefits [6–9].

Hard oxidation is known as hard anodizing (HA) treatment, in which a metal is placed in an electrolyte as the anode such that an oxide film of thickness ranging from tens to hundreds of microns is formed on the metal surface. The formation of such an oxide

layer film endows the metal with corrosion and wear-resistant properties, but it also has the disadvantage of low hardness [10]. Microarc oxidation (MAO) is a surface treatment technology developed from anodic oxidation. It increases the electrode voltage of common anodic oxidation from the Faraday zone to the high voltage discharge zone, produces microarc plasma spark discharge, and in situ forms a ceramic coating on the surface of valve metals such as Al, Mg, Ti, and their alloys [11–13]. Diamond-like carbon coating is a substable amorphous carbon film combining $sp^3$-hybridized bonds (diamond structure) and $sp^2$-hybridized bonds (graphite structure), which is considered to be an ideal surface-protective coating to improve the performance and life of aluminum alloys due to its high hardness, excellent abrasion resistance, low coefficient of friction, high elastic modulus, and good chemical inertness [14–21]. Depending on its structure, it can be divided into hydrogen-containing DLC coatings and hydrogen-free tetrahedral amorphous carbon (ta-C) coatings. Many researchers have studied these three coating preparation methods and found that the hard anodic oxide layer has a significant influence on the wear behavior of aluminum alloy. Soffritti et al. [22] prepared anodic aluminum oxide coatings with different thicknesses by different hard oxidation methods and observed their microstructure and mechanical properties. Through the wear test, it was concluded that the anodic aluminum oxide coatings have a great influence on the wear performance of aluminum alloys. P Kwolek et al. [23] prepared hard oxide coatings on 5005 and 6061 aluminum alloys. The wear resistance of hard oxide coatings prepared on different aluminum alloys was studied by a scratch test and a wear test. The results show that the wear resistance of coated 6061 aluminum alloy is higher than that of 5005 aluminum alloy, which is mainly related to the lower porosity and higher hardness of hard oxide coatings prepared on 6061 aluminum alloy. In addition, the wear resistance of MAO coatings has also been studied. For example, J.J. Zhuang et al. [24] formed MAO coatings on aluminum alloys and studied the effects of oxide films formed at different oxidation times on their wear resistance. C. Yang et al. [25] prepared MAO coating on aluminum alloy by changing the concentration of phosphate, which increased the hardness of the coating and reduced the wear rate. Furthermore, some studies were carried out on the effects of DLC coatings on the surface of aluminum alloys [26,27]. It was found that the hardness was significantly higher than that of uncoated aluminum alloys, but there is a high residual stress and poor adhesion between the DLC film and the substrate. The surface properties can be optimized by doping elements. It has been reported that metal elements (such as Cr [28], Ti [29], W [30], Zr [31], Ni [32], and Cu [33]) can effectively release residual stress by changing the structure of the DLC film. Therefore, the above three surface modification coatings can effectively improve the hardness and wear resistance of aluminum alloy, but comparisons of these three types of coated aluminum alloy are scarce, so this aspect is worth studying.

In this study, three types of coatings, named MAO, HA+W+DLC, and HA+Ti+ta-C, were deposited on 7A04 aluminum alloy and compared with an aluminum alloy substrate without surface treatment. The microstructure, hardness, and wear behavior of these three coatings were investigated.

## 2. Test Materials and Methods

### 2.1. Test Material

In this study, a 7A04 aluminum alloy cuboid sample with a size of 19 mm × 12 mm × 12 mm was used as the test matrix material. A 25Cr3Mo3NiNbZr die steel ring with an inner diameter of 42 mm, an outer diameter of 50 mm, and a width of 13 mm was used as the grinding material. The nominal compositions of the materials are listed in Tables 1 and 2.

**Table 1.** Chemical composition of 7A04 aluminum alloy (wt.%).

| Element | Si | Fe | Cu | Mn | Mg | Cr | Zn | Ti | Al |
|---|---|---|---|---|---|---|---|---|---|
| content | 0.5 | 0.5 | 1.4–2.0 | 0.2–0.6 | 1.8–2.8 | 0.1–0.25 | 5.0–7.0 | 0.1 | Balance |

**Table 2.** Chemical composition of 25Cr3Mo3NiNbZr steel (wt.%).

| Element | C | Si | Mn | Cr | Mo | Ni | Nb | Zr | V |
|---------|------|------|------|------|------|------|------|--------|---------|
| content | 0.28 | <0.1 | 0.18 | 3.03 | 2.94 | 0.55 | 0.14 | 0.0012 | Balance |

### 2.2. Test Method

Prior to coating deposition, the 7A04 aluminum alloy substrates were polished with 2000 mesh sandpaper. Microarc oxidation was carried out using an electrolyte containing 30 g/L $(NaPO_3)_6$, 3 g/L KOH, 8 g/L $Na_2B_4O_7$, and 15 g/L glycerin for 35 min under a current density of 10 A·dm$^{-2}$. The temperature of the electrolyte was maintained at 303 K, while the sample was placed as an anode, and the stainless steel was used as the cathode. When preparing multilayer diamond-like carbon coatings, the Cr/Ni-doped oxide film transition layer was firstly coated on the surface of aluminum alloy by hard oxidation, which was conducted using a 15 wt.% sulfuric acid electrolyte for 30 min under a current density of 50 mA·cm$^{-2}$, and then conducted by repetition of dipping in 3 M Cr/Ni-containing electrolyte for 1 min and heat treatment in a 400 °C electric furnace for 30 min. Secondly, the W/Ti transition layer was prepared using a nonequilibrium magnetron sputtering system (UDP-650, Teer Coatings Co., Ltd., Droitwich, UK) with a W/Ti target separately, under the conditions of a constant current of 1 A and a negative bias voltage of −60 V for 5 min to enhance the bonding strength. Finally, the DLC layer was prepared by magnetron sputtering under the conditions of a constant current of 3.5 A and a negative bias voltage of −60 V for 150 min. The ta-C layer was prepared by arc ion plating with optimized parameters of a current of 60 A, a bias voltage of −120 V, and a depositing time of 60 min.

Friction and wear tests were carried out for 4000 cycles over 1200 s using a high-speed ring-block friction meter (MRH-3, Jinan Shunmao Corporation Ltd., Jinan, China) under oil lubrication with a load of 300 N and a speed of 200 rpm/min. The test was carried out at room temperature and atmospheric pressure. Bulk samples were prepared from uncoated and differently coated 7A04 aluminum alloys. The ring specimens were made of DLC-coated 25Cr3Mo3NiNbZr mold steel. After the test, the samples were firstly shaken ultrasonically for 10 min using petroleum ether, which acted as the organic solvent to remove the surface lubricating oil, and then were ultrasonic cleaned in anhydrous ethanol for 10 min and dried at 60 °C for 0.5 h. After drying, the samples were weighed immediately (five times for each measurement). The maximum and minimum values were removed, and the remaining average value was considered. The weighing method was the same as that used after the testing.

The wear surface morphologies were analyzed by a laser confocal microscope (OLS5100, Reco System Integration Ltd., Beijing, China). A field-emission scanning electron microscope (JSM-IT800, JEOL Companies, Tokyo, Japan) was used to observe the surface and cross-sectional morphology before and after wear and to analyze the characteristics of friction wear and wear behavior. It was combined with an SEM-supporting energy spectrometer to perform energy spectral analysis. The X-ray diffractometer (Brux-D8, Bruker AXS Companies, Karlsruhe, Germany) was used to analyze the phase composition of the worn specimen surface using Cu-Kα radiation at 40 kV and 40 mA. Cu was used as the target material for the tests, and the scanning range was set from 20° to 80° with a scanning speed of 6°/min. In these experiments, a nanoindentation tester (Keysight G200, Keysight Technologies, Santa Rosa, CA, USA) was chosen to measure the coating hardness using the continuous stiffness method and the 10% coating thickness method, with a point-to-point spacing of 20 μm, a load setting of 50 mN, and a Poisson's ratio of 0.26. The bonding states of carbon atoms in two multilayer coatings were measured by Raman spectroscopy (InVia, Renishaw, Lundon, UK).

## 3. Results and Discussion

### 3.1. Pre-Wear Microscopic Morphology Analysis

The surface morphologies of different coatings before wear are shown in Figure 1. As shown in Figure 1a,b, the MAO-coated specimens have an inhomogeneous surface and are endowed with a typical "crater" porous structure constituting several micropores, a few small particles, and microcracks. Micropores are formed by molten oxides and bubbles discharged from the microarc discharge channels, through which molten alumina flows out and solidifies rapidly. Small particles also solidify from these molten oxides. Simultaneously, a large amount of the anionic component $PO_3^-$ and a small amount of $B_4O_7^{2-}$ in the electrolyte enter the channel, leaving a clear and distinct boundary. The cracks are mainly caused by the different coefficients of phase expansion in the coatings and thermal stresses [34]. As shown in Figure 1c–f, continuous dense diamond-like coatings are successfully deposited on the surface of the aluminum alloy, and both coating surfaces appear smooth and flat owing to the dense arrangement of small amorphous particles. The HA+W+DLC coating has more cracks than the HA+Ti+ta-C coating, which has a smoother surface topography. The surface morphologies of the substrate material shown in Figure 1g,h are not flat and have stray scratches and cracks related to the preparation of the specimens for machining. The cross-sectional morphologies of the investigated coatings and the corresponding elemental line scans are shown in Figure 2. The thickness of the as-deposited MAO coating, HA+W+DLC coating, and HA+Ti+ta-C coating is 37.1 μm, 9.8 μm, and 3.2 μm, respectively. It can be seen that the three coatings are dense with no pores and well bonded with the substrate, and the interfaces of each sublayer in the two multilayer coatings are clear. Additionally, the distribution of Al and O elements is relatively uniform in MAO, and the composition of HA+W+DLC and HA+Ti+ta-C conforms to the design, which is conducive to binding with the substrate.

### 3.2. XRD Patterns

The X-ray diffraction patterns of the coating and substrate after abrasion are shown in Figure 3. The 7A04 aluminum alloy substrate shows a strong aluminum peak, mainly composed of the aluminum phase, while the MAO coating also shows an aluminum phase, owing to the high porosity [35] and low thickness of the coating. The MAO coating is mainly composed of $\alpha$-$Al_2O_3$ and $\gamma$-$Al_2O_3$. The diffraction peak of $\gamma$-$Al_2O_3$ is stronger than that of $\alpha$-$Al_2O_3$, indicating that the oxide of the MAO coating is mainly composed of $\gamma$-$Al_2O_3$. The HA+W+DLC and HA+Ti+ta-C coatings are very thin, and the rays can penetrate the coating directly, showing the aluminum phase corresponding to the substrate.

### 3.3. Hardness Analysis

The hardness of the MAO, HA+W+DLC, HA+Ti+ta-C, and the substrate was measured using a nanoindentation instrument. Using the continuous stiffness and 10% coating thickness methods, five indents were created in each sample; the average value of the maximum and minimum values of the five indentation removals was considered to be the surface hardness of the specimen. Finally, the average hardness of each coating specimen was calculated, as shown in Figure 4. The average hardness of the 7A04 aluminum alloy substrate was 1.74 GPa, and the average hardness of the MAO coating was 7.33 GPa, which was four times higher than that of the substrate. The average hardness of the HA+W+DLC coating after hard anodizing treatment was 22.64 GPa, which was 13 times higher than that of the substrate. The average hardness of the HA+Ti+ta-C coating after hard anodizing treatment was 34.23 GPa, which was 19 times higher than that of the substrate. It was found that the substrate and MAO coating had a better bonding force, which was beneficial to the performance of coating hardness. The hardness of the HA+W+DLC and HA+Ti+ta-C coatings after hard anodizing treatment considerably improved owing to the typical diamond-like structure within the HA+W+DLC coating.

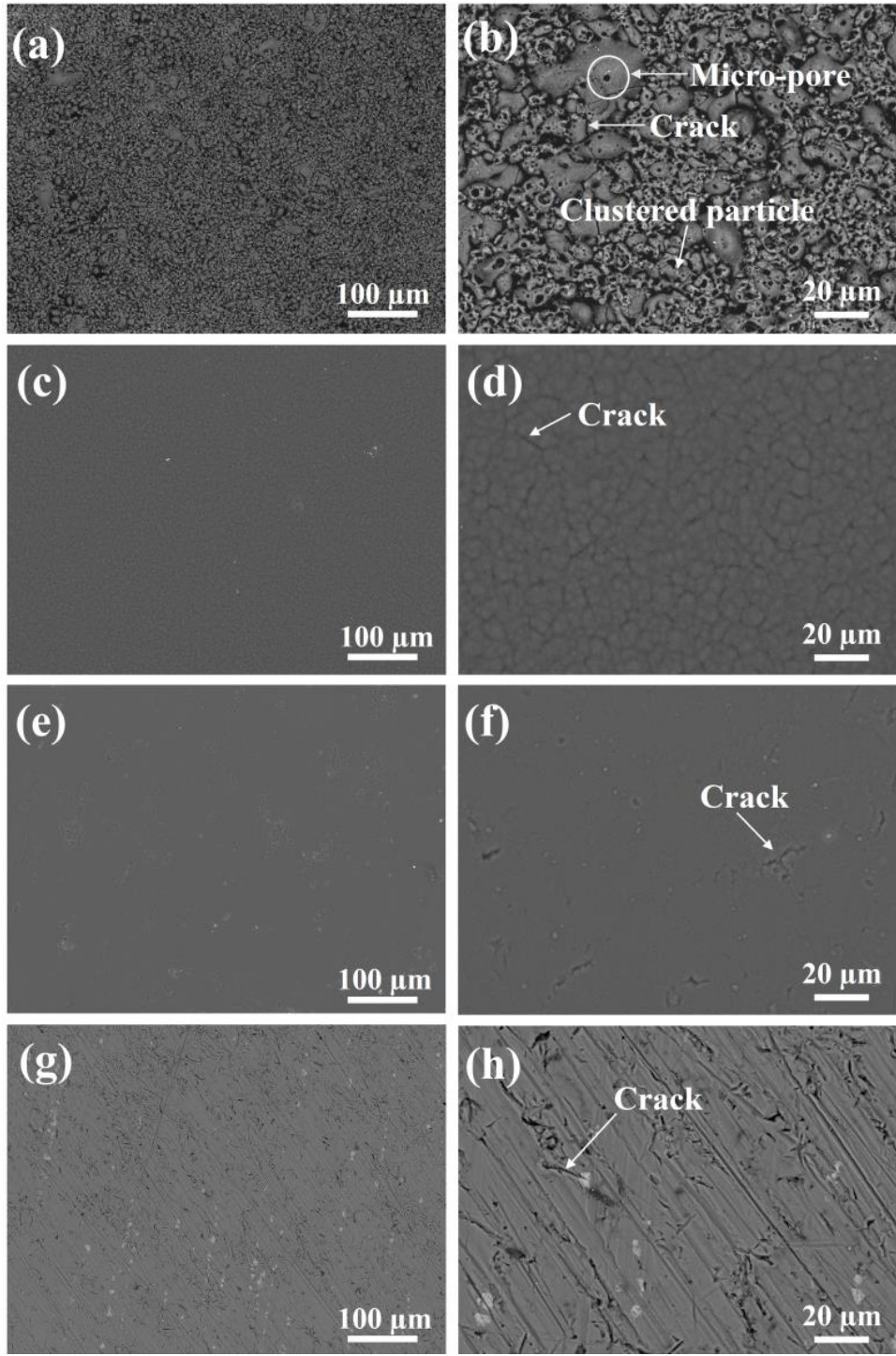

**Figure 1.** Surface morphologies of (**a**) MAO coating (×200 times), (**b**) MAO coating (×800 times), (**c**) HA+W+DLC coating (×200 times), (**d**) HA+W+DLC coating (×800 times), (**e**) HA+Ti+ta-C coating (×200 times), (**f**) HA+Ti+ta-C coating (×800 times), (**g**) 7A04 substrate (×200 times), and (**h**) 7A04 substrate (×800 times).

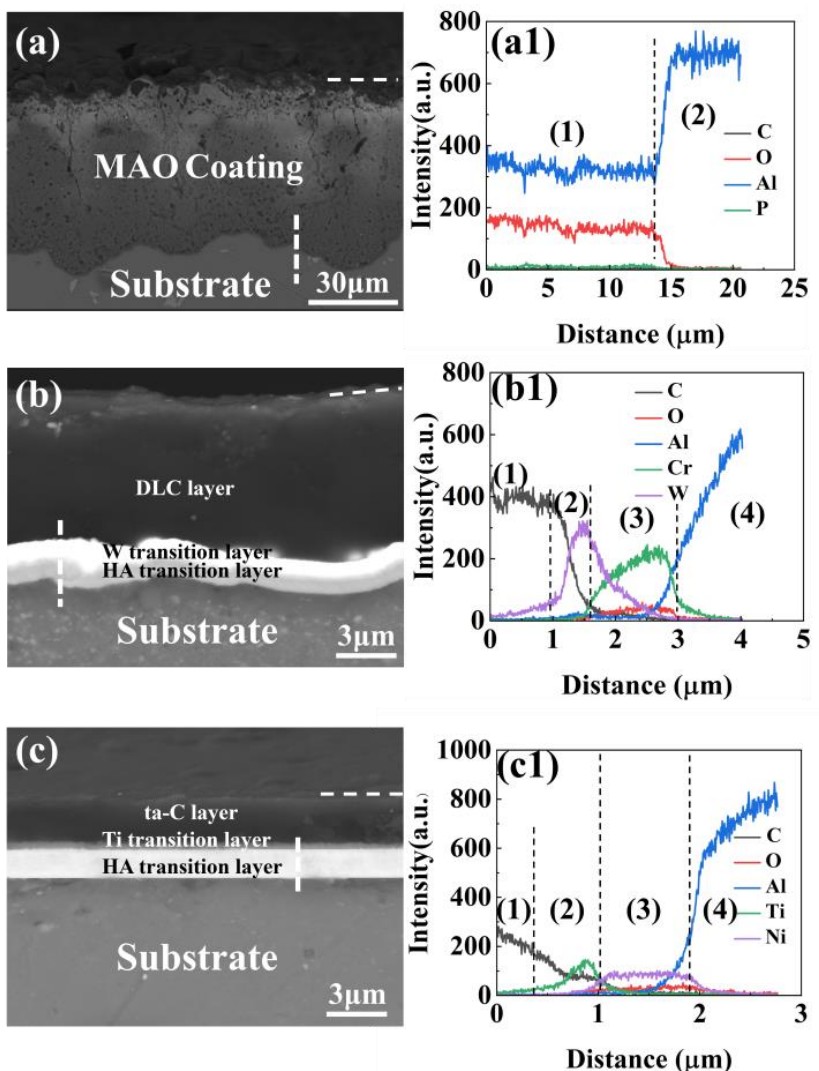

**Figure 2.** Cross-section morphologies and the corresponding elemental line scans of (**a**) MAO coating, (**b**) HA+W+DLC coating, and (**c**) HA+Ti+ta-C coating.

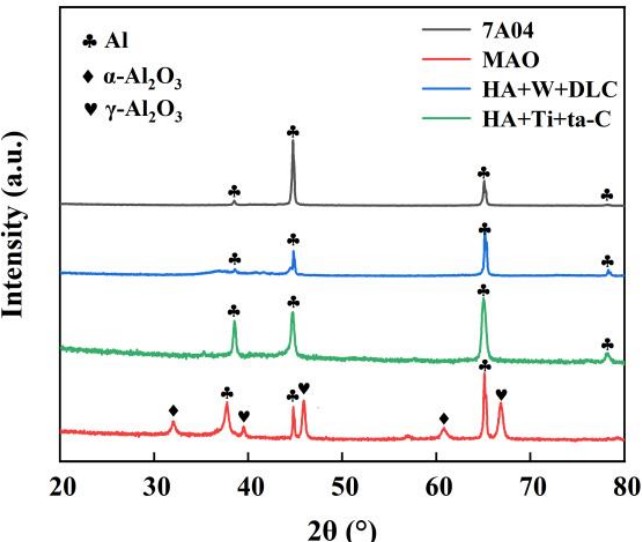

**Figure 3.** XRD patterns of MAO, HA+W+DLC, HA+Ti+ta-C, and 7A04 substrate.

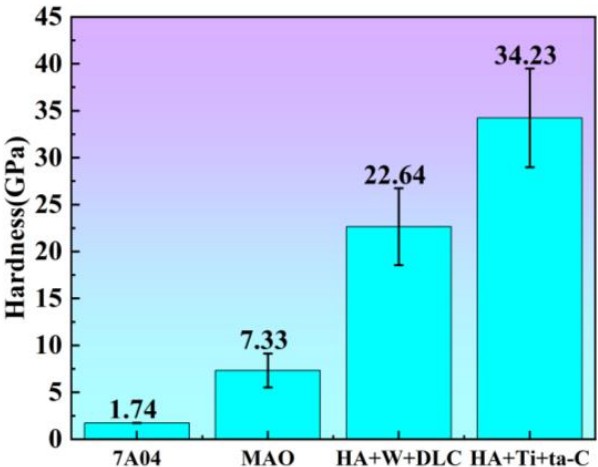

**Figure 4.** Hardness diagrams of MAO, HA+W+DLC, HA+Ti+ta-C, and 7A04 substrate.

*3.4. Wear Test*

3.4.1. Friction Coefficient and Wear Rate

The coefficient of friction of different coatings against the substrate as a function of the wear time is shown in Figure 5. The wear process is typically divided into three stages—initial, break-in, and stable wear [36]. As shown in Figure 5a, in the initial stage (0–10 s), due to its rough, porous structure, the tangential resistance is large during wear, and the friction coefficient gradually increases. It can be preliminarily determined that the coating begins to break, gradually peeling off. The loose, porous shape of the exterior surface stores the abrasive particles generated by friction. Subsequently, the abrasive particles form a relatively smooth protective coating on the surface, leading to a decrease in the coefficient of friction, which stabilizes at 0.03 [37]. As evident in Figure 5b,c, for the HA+W+DLC and HA+Ti+ta-C coatings, their average friction coefficients of 0.028 and 0.025, respectively, are very low after the break-in period. The average friction coefficients of the HA+Ti+ta-C coatings are slightly less than those of the HA+W+DLC coatings. It can be determined that the friction coefficient gradually fluctuates with an increase in the wear time, but there is no obvious "transition point" in the three coatings, indicating that the coating still exists [38,39]. In Figure 5d, the friction coefficient curves of the three coatings are compared with those of the 7A04 aluminum alloy substrate. The friction coefficient of the substrate is between 0.01 and 0.04, and it fluctuates considerably. In the initial stage of wear, the coefficient first increases rapidly and then decreases, possibly related to the accelerated oxidation of the frictional heat generated during the wear process. For the untreated substrate material, there is no protective coating on the surface, and the rapid fluctuation in the friction coefficient within 90 s indicates that the steel ring undergoes serious wear on the surface of the aluminum alloy. The combined extent of wear and wear rate are shown in Figure 5e,f. All three coatings reduce the mass loss and wear rate compared to the 7A04 substrate. It can be seen that all three coatings prepared on the 7A04 surface have a good anti-wear effect, resulting in an improved wear resistance of the substrate. The lower mass loss and wear rate of the HA+W+DLC coating and the HA+Ti+ta-C coating after hard anodizing treatment are $1.36 \times 10^{-7}$ mm$^3$/N·m and $1.51 \times 10^{-7}$ mm$^3$/N·m, respectively. Mainly due to the typical amorphous material after hard anodizing treatment [40], HA+W+DLC and HA+Ti+ta-C have both sp$^3$- and sp$^2$-type bonding structures. The vibration mode of sp$^2$ represents the characteristics of graphite and hence has some characteristics of graphite, such as lubrication. Hence, the MAO, HA+W+DLC, and HA+Ti+ta-C coatings are slightly abraded, and the wear rate of HA+Ti+ta-C is lower than that of HA+W+DLC [41]. All coatings exhibit slight wear, with HA+Ti+ta-C showing a lower wear rate than HA+W+DLC, followed by MAO.

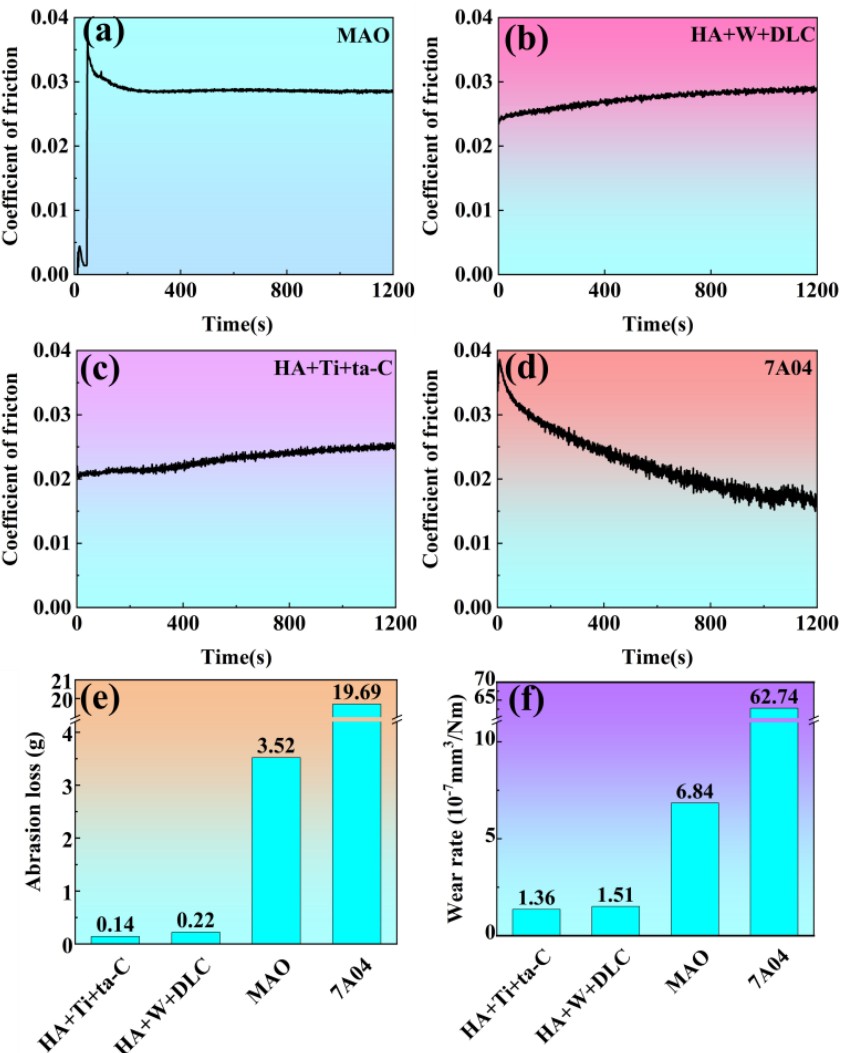

**Figure 5.** (**a–d**) Coefficient of friction of MAO coating, HA+W+DLC coating, HA+Ti+ta-C coating, and 7A04 substrate and corresponding (**e**) abrasion loss and (**f**) wear rate.

### 3.4.2. Three-Dimensional Morphology

The surface morphologies of the substrate and coating were observed using a three-dimensional profiler. The three-dimensional morphology of the three coatings and their surfaces are shown in Figure 6. It is evident that compared with the wear depth of 179.05 μm of the substrate material (Figure 6d,d1), the wear trace of the HA+Ti+ta-C coating is the shallowest, at 13.95 μm (Figure 6c,c1), followed by that of the HA+W+DLC coating after hard anodizing treatment (18.55 μm, as shown in Figure 6b,b1), and the wear depth of the MAO coating (35.33 μm, as shown in Figure 6a,a1). Combined with the adhesion between the coating and substrate, the wear of the HA+W+DLC and HA+Ti+ta-C coatings is the least, which is related to their having the lowest-roughness substrate, the preparation method, and the thickness of the coating. The results show that compared with the wear degree of the 7A04 aluminum alloy substrate, the wear resistance of the other three coated aluminum alloys is improved. Among them, the wear depth of HA+Ti+ta-C is the shallowest, which is consistent with the law of the friction coefficient wear rate, indicating better wear resistance.

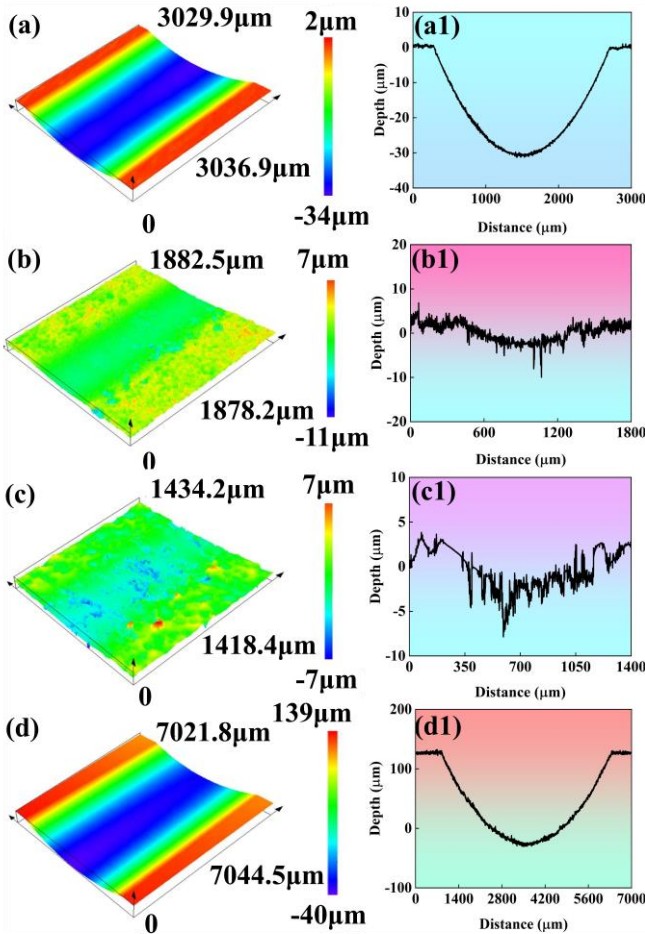

**Figure 6.** Three-dimensional topographies and height profiles of (**a**,**a1**) MAO coating, (**b**,**b1**) HA+W+DLC coating, (**c**,**c1**) HA+Ti+ta-C coating, and (**d**,**d1**) 7A04 substrate.

3.4.3. Wear Surface Morphology Analysis

The surface morphologies of the four kinds of samples with the worn part and the unworn part are shown in Figure 7. It can be seen from Figure 7a that the wear mechanism of the MAO coating is mainly abrasive wear. This is due to the friction between the MAO coating and the friction pair after wear. The friction pair first contacts the convex part of the MAO coating to form a nonuniform contact, and the actual contact occurs at the contact point. Under a certain load and speed, some microprotrusions are worn off, and wear debris is formed on the worn track. In the subsequent friction and wear test, microcutting occurs to accelerate the wear of the MAO coating, indicating that the wear mechanism is abrasive wear. After a period of wear, the surface of the microprotruding becomes flat, which increases the contact area between the MAO surface and the friction pair. When the load is greater than the strength of the microconvex part on the MAO coating, the convex part cracks to form a large abrasive particle. Under the cyclic action of the contact stress between the MAO and the friction pair, the crack propagates along the pore edge of the MAO coating, resulting in the separation of the MAO through abrasive wear. Combined with the EDS element distribution, it can be seen that the distribution of O, Al, and P elements is relatively uniform, and there is no significant difference. It can also be proved that the MAO coating is not completely destroyed. As shown in Figure 7b,c, the HA+W+DLC and HA+Ti+ta-C coatings have flake morphology. It can be seen that wear of the coating occurs on the surface of HA+W+DLC and HA+Ti+ta-C, accompanied by abrasive wear on the substrate due to debris generated after wear. According to the distribution of EDS elements, the surface of HA+W+DLC coating has Cr, W, C, O, and Al, and the surface of HA+Ti+ta-C coating has Ti, Ni, C, O, and Al. Among them, Cr, W, Ti,

and Ni are doped elements, which are used to reduce problems regarding the high internal stress and poor thermal stability of the coating and substrate [42]. As shown in Figure 7d, it can be seen that there are deep parallel furrows in the aluminum alloy matrix after the wear test, and obvious abrasive wear can be seen.

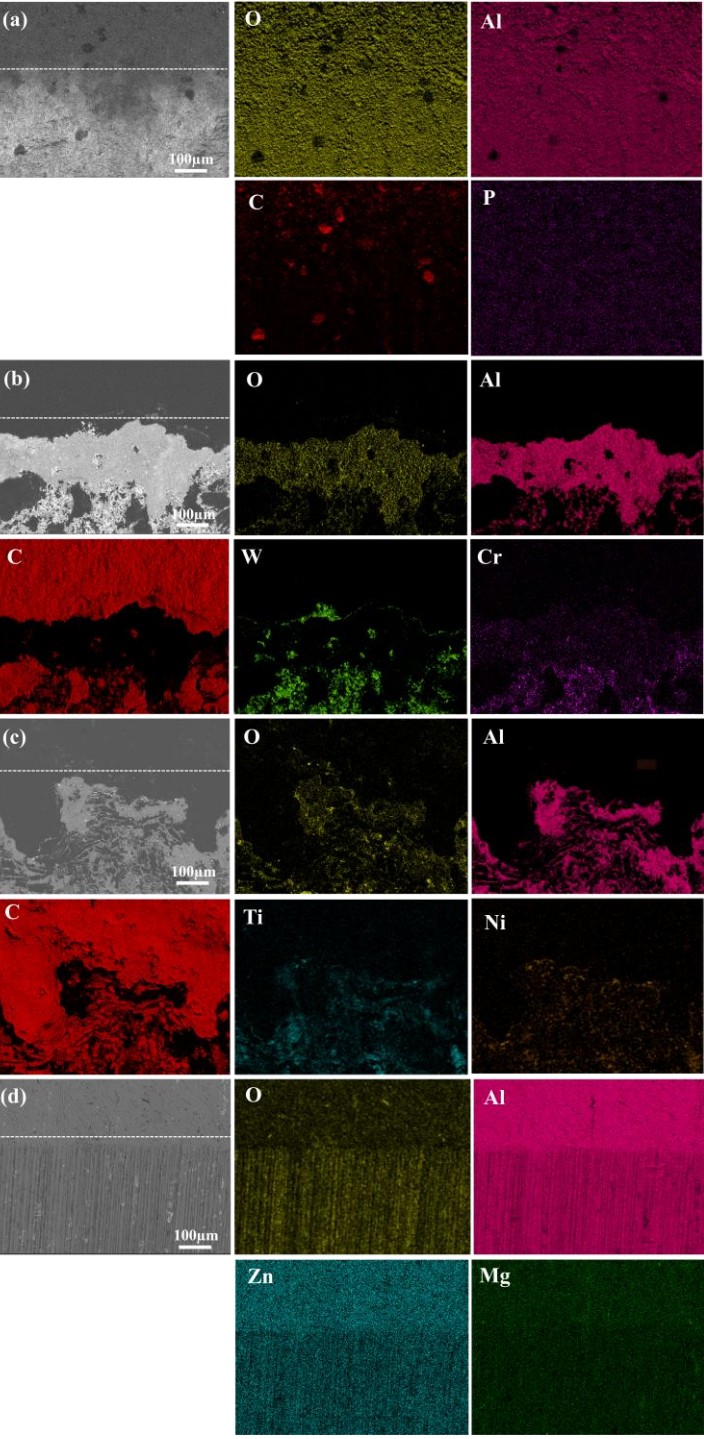

**Figure 7.** SEM surface morphologies and element distribution maps of (**a**) MAO coating, (**b**) HA+W+DLC coating, (**c**) HA+Ti+ta-C SEM, and (**d**) 7A04 substrate.

The SEM cross-sectional morphologies and element distributions of the different samples are shown in Figure 8. It can be seen from Figure 8a that the thickness of the MAO coating after wear is approximately 32 μm, and no connected pores are observed at the

cross-section of the MAO coating. Furthermore, it does not extend to the substrate, which indicates that the bonding strength between the MAO coating and the substrate is high [24]. The endpoint of the diffusion of electrolyte to the substrate is between the MAO coating and substrate, which is the starting point of the microarc discharge channel [43]. The high temperature and pressure induced by the microarc discharge can promote the diffusion of $O^{2-}$ and $Al^{3+}$ in the region to form molten Al-O compounds, simultaneously transferring heat to the substrate to melt Al. The molten Al-O and Al undergo chemical microalloying reactions under the effects of thermochemistry, electrochemistry, and plasma chemistry. As shown in the line scan in Figure 8a1, P, O, and C diffuse from the electrolyte; Al and Mg diffuse from the matrix; the concentrations of Al and O are the highest; and the transition at the bonding interface is evident. This indicates that in the MAO reaction, aluminum mainly diffuses from the inside to the outside, and the electrolyte may also provide a small amount. This indirectly indicates that the MAO coating is generated in situ on the surface of the 7A04 aluminum alloy. The bonding at the interface is based on the metallurgical bonding of chemical bonding and diffusion, which prevent further separation of the coating layer and effectively reduce wear. From Figure 8b,b1,c,c1, it is evident that the thickness of the HA+W+DLC and HA+Ti+ta-C coatings after wear is approximately 9 μm and 3 μm, respectively. In the HA+W+DLC coating, Cr and W are the doping elements, W forms a transition layer, and Cr is hard-oxidized to oxide. Ti and Ni are the doping elements of the HA+Ti+ta-C coating, Ti forms a transition layer, and Ni is hard oxidized to oxide. The whole coating structure is dense and uniform without obvious pores or cracks, and there is a clear boundary between the substrate, the transition layer, and the coating.

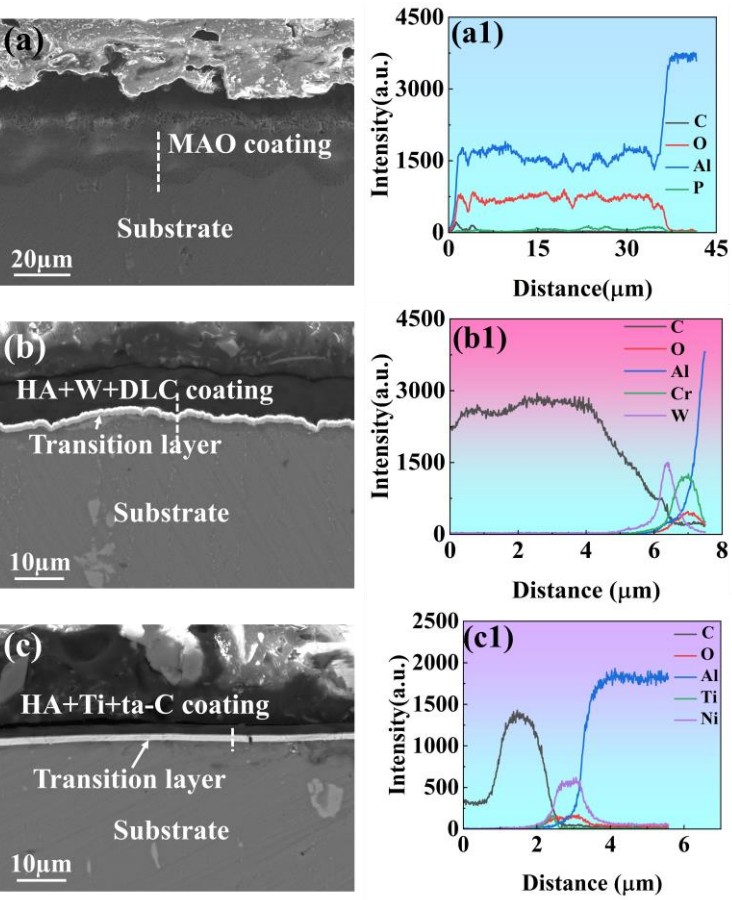

**Figure 8.** SEM cross-sectional morphologies and element distributions of (**a**,**a1**) MAO coating, (**b**,**b1**) HA+W+DLC coating, and (**c**,**c1**) HA+Ti+ta-C coating.

### 3.4.4. Raman Spectroscopy

Raman spectroscopy is a reliable method of analyzing the microstructure of a DLC coating and characterizing $sp^2$ and $sp^3$ C-C hetero-bonds [44,45]. The DLC coatings demonstrated a broad diffuse peak at 1200–1700 cm$^{-1}$ and a weak shoulder peak at 1300–1400 cm$^{-1}$. Two Gauss peaks were obtained by fitting the 'D' peak near 1332 cm$^{-1}$ (characterizing the $sp^3$ C-C hetero-bonds) and the 'G' peak near 1575 cm$^{-1}$ (characterizing the $sp^2$ C-C hetero-bonds) with the Gauss function. These represent the characteristic Raman peaks of diamond and graphite, respectively.

The Raman spectra of the HA+W+DLC and HA+Ti+ta-C coatings are shown in Figure 9. The laser wavelength was 532 nm, and the wavelength range was 200–2000 nm$^{-1}$. Figure 9a,b show the Raman spectra of two typical DLC coatings fitted using the original software. The two DLC coatings prepared in this study exhibit typical Raman characteristics. The intensity ratio of the D peak to the G peak, ID/IG, was proportional to the number ratio of $sp^2/sp^3$ C-C bonds [46]. The ID/IG values of the HA+W+DLC and HA+Ti+ta-C coatings after hard anodizing treatment, calculated using Origin software(2022), were 1.11 and 0.89, respectively. Thus, the highest content of $sp^3$ C-C bonds was in the HA+Ti+ta-C coating after hard anodizing treatment, followed by the HA+W+DLC coating. It is known from the Raman spectrum that the D peak represents the mixed vibration mode of the $sp^2$, or $sp^2$ and $sp^3$, bond structure in the coating. The higher the content of the $sp^3$ bond structure in the system, the more prevalent the tetrahedral structure in the coating, the more the structure is biased toward the diamond structure, and the greater the hardness of the coating. A comparison of the Raman spectra of the two coatings shows that the HA+Ti+ta-C coatings had a higher $sp^3$ bond content, which is consistent with these results.

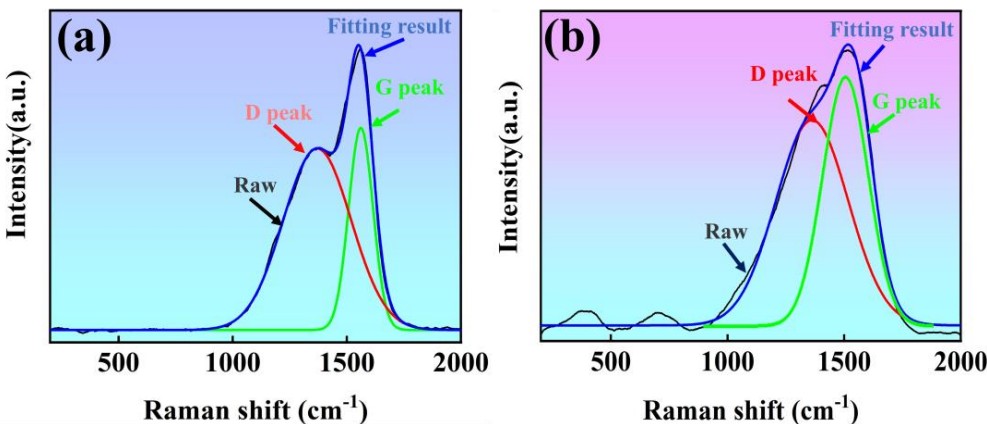

**Figure 9.** Raman spectra of (**a**) HA+W+DLC and (**b**) HA+Ti+ta-C.

### 4. Conclusions

(1) In this work, MAO, HA+W+DLC, and HA+Ti+ta-C coatings were successfully prepared on a 7A04 aluminum alloy substrate. The surface hardness was significantly improved after coating. The hardnesses of three coated samples were 7.33 GPa, 22.64 GPa, and 34.23 GPa, respectively. The highest hardness of the HA+Ti+ta-C coatings resulted from the high $sp^3$ C-C bond content.

(2) During the ring-block wear tests under oil lubrication, both multilayer coatings exhibited excellent wear resistance. The average coefficient of friction and wear rate of HA+W+DLC and HA+Ti+ta-C were, respectively, 0.028 and $1.51 \times 10^{-7}$ mm$^3$/N·m, and 0.025 and $1.36 \times 10^{-7}$ mm$^3$/N·m. The higher surface hardness of the HA+Ti+ta-C coating led to better wear resistance, which suggests that the coating can be applied in the surface protection of aluminum alloys.

**Author Contributions:** Conceptualization, Methodology, Investigation, and Data Curation, J.H. and C.Z.; Writing—Original Draft, J.H.; Software, X.W. and X.M.; Conceptualization and Writing—Review

and Editing, C.D., H.Y. and C.W.; Methodology, Study Design, and Writing—Review and Editing, J.X., Z.Q. and T.J. All authors have read and agreed to the published version of the manuscript.

**Funding:** This project was supported by the State Key Lab of Advanced Metals and Materials (2022-Z17), Frontier Exploration Projects of Longmen Laboratory (NO. LMQYTSKT011), and Scientific and Technological Project of Henan Province (222102230033).

**Institutional Review Board Statement:** Not applicable.

**Informed Consent Statement:** Not applicable.

**Data Availability Statement:** Data are contained within the article.

**Acknowledgments:** We wish to take this opportunity to thank the Provincial and Ministerial Co-construction of Collaborative Innovation Center for Non-ferrous Metal New Materials and Advanced Processing Technology for their support.

**Conflicts of Interest:** Tao Jiang was employed by the company Luoyang Wanji Aluminum Processing Co., Ltd., Ziping Qiao and Jun Xue was employed by the company Science and Technology on Transit Impact Laboratory, No.208 Research Institute of China Ordnance Industries. The remaining authors declare that the research was conducted in the absence of any commercial or financial relationships that could be construed as a potential conflict of interest.

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
