# Peer review of "Improving the Wear Resistance Properties of 7A04 Aluminum Alloy with Three Surface Modification Coatings"

_coatings, doi:10.3390/coatings14040476_

Round 1
Reviewer 1 Report
Comments and Suggestions for Authors
Review on the Manuscript entitled:
Improving the mechanical properties of 7A04 aluminum alloy 2 by three surface modification coatings
Dear Editor,
The authors have investigated three surface modification coatings were successfully prepared on 15 the surface of 7A04 aluminum alloy by micro-arc oxidation (MAO) and a combination of hard ano-16 dizing treatment (HA) and physical vapor deposition (PVD), named MAO, HA+W+DLC, and 17 HA+Ti+ta-C,.…The results have indicated that the surface quality 19 and hardness of coated samples are higher than those of the 7A04 aluminum alloy, and the 20 HA+Ti+ta-C coating possesses the highest hardness of 34.23 GPa. In my opinion, the subject of this manuscript is interesting and applicable for other researchers. I recommend this article for publishing in the Coatings; however, it needs some major revisions:
1. The different parts of Abstract including purposes, methods, results, and conclusions have been well designed, but I recommend the authors to indicate the main reason of choosing Al alloy by one or two sentences that can attract readers attention.
2. Page 3, Lines 98 & 99: “The maximum and minimum values were removed, and the remaining average value was considered”. Please describe the maximum, minimum and average values.
3. Page 3, Lines 105 & 106: “It was combined with an SEM-supporting energy spectrometer to perform energy spectral analysis”. Please remake the value of SEM analysis in a Table or with a graph.
4. Page 3, Lines 122 & 123: “The cracks are mainly caused by the different coefficients of phase expansion in the coatings and thermal stresses [34]”. Have the authors applied graphene particles like Ref 34? Please explain it.
5. Page 4, lines 148 &149: “It is analyzed that the substrate and MAO coating have a better bonding force, which in-148 creases the hardness”. Is there any probability of encountering chemisorption status. If yes, which kind of adsorption can be estimated? Please explain it.
6. Page 5, line 151: Where are Raman spectra? In which frequencies and intensities? Please explain it.
7. Please increase the quality of Figure 3. The parameters in the x and y axes are not clear. What is the total consequence of Figure 3 regarding the similarities and differences of the graphs? Please describe it.
8. Is titanium (Ti) as a light transition element can be an appropriate option as the additive to aluminum (Al) alloy?
9. Please increase the quality of Figure 4. The parameters in the x and y axes are not clear.
10. Page 8, lines 224 & 225: “Combined with the EDS elemental distribution, it is evident that the distribution of O, Al, and P is relatively uniform, and there is no significant difference”. However, the properties of these elements are different. How can the authors illustrate it?
11. The experimental conditions including temperature and pressure have not been discussed.
12. The authors have employed a variety of metal and metalloids atoms with different physical and chemical characteristics. Therefore, please mention the limitations of your work.

Author Response
Corrections made for Coatings-2929440
Original Title: Improving the mechanical properties of 7A04 aluminum alloy by three surface modification coatings
Dear editor and reviewer,
We appreciate the helpful comments of the reviewers. The manuscript has been carefully revised in light of these comments, and we hope that the improved manuscript is acceptable for publication. Every change was outlined in the red color in the revised manuscript. The responses to comments are listed as follows (The reviewer’s comments are in italic).
- The different parts of Abstract including purposes, methods, results, and conclusions have been well designed, but I recommend the authors to indicate the main reason of choosing Al alloy by one or two sentences that can attract readers attention.
Thank you for your friendly suggestion. We have modified the Abstract in the new manuscript, please see Lines14-16, Page 1.
- Page 3, Lines 98 & 99: “The maximum and minimum values were removed, and the remaining average value was considered”. Please describe the maximum, minimum and average values.
Thanks for the comment. The maximum, minimum, and average values of the mass losses are shown in Table S1.
Table S1 Mass losses of the different samples
|
|
Mass loss (g) |
|||
|
7A04 |
MAO |
HA+W+DLC |
HA+Ti+ta-C |
|
|
1 |
20.30 |
3.49 |
0.24 |
0.14 |
|
2 |
19.10 |
3.53 |
0.20 |
0.15 |
|
3 |
19.40 |
3.54 |
0.22 |
0.13 |
|
Max |
21.70 |
3.55 |
0.25 |
0.21 |
|
Min |
18.90 |
3.48 |
0.20 |
0.11 |
|
Average |
19.60 |
3.52 |
0.22 |
0.14 |
- Page 3, Lines 105 & 106: “It was combined with an SEM-supporting energy spectrometer to perform energy spectral analysis”. Please remake the value of SEM analysis in a Table or with a graph.
Thanks for your suggestion. In this work, EDS was used for the element mapping distribution of the worn sample surfaces (Fig. 7) and the line scanning of the worn sample sections (Fig. 8) respectively. The element contents of different samples in mapping are listed in the following Table S2.
Table S2 Elemental mapping distributions of different samples
|
Elements |
Content (wt.%) |
|||
|
MAO |
HA+W+DLC |
HA+Ti+ta-C |
7A04 |
|
|
C |
21.27 |
76.76 |
84.71 |
|
|
O |
12.57 |
7.67 |
4.57 |
10.62 |
|
Al |
35.62 |
12.04 |
8.74 |
81.50 |
|
P |
0.54 |
|
|
|
|
Cr |
|
0.38 |
|
|
|
W |
|
3.14 |
|
|
|
Ti |
|
|
1.08 |
|
|
Ni |
|
|
0.90 |
|
|
Mg |
|
|
|
2.02 |
|
Zn |
|
|
|
5.86 |
- Page 3, Lines 122 & 123: “The cracks are mainly caused by the different coefficients of phase expansion in the coatings and thermal stresses [34]”. Have the authors applied graphene particles like Ref 34? Please explain it.
Thanks for the comments. We did not apply graphene particles in this work. We cite this paper here to illustrate similar cracking phenomena caused by thermal stress due to different coefficients of thermal expansion, and perhaps later we will try to add graphene particles during the micro-arc oxidation process and study the performance of this coating.
- Page 4, lines 148 &149: “It is analyzed that the substrate and MAO coating have a better bonding force, which increases the hardness”. Is there any probability of encountering chemisorption status. If yes, which kind of adsorption can be estimated? Please explain it.
Thanks for the comment. Micro-arc oxidation is a process of in-situ growth of ceramic films, dominated by matrix metal oxides, under instantaneous high temperature and high pressure caused by arc discharge. During this process, oxygen is adsorbed on the aluminum alloy by virtue of oxide formation (Al2O3). Therefore, there is certainly encountering chemisorption status. This chemical adsorption state helps to make the bonding between the oxidation coating and the aluminum alloy substrate stronger and enhances the adhesion of the coating.
- Page 5, line 151: Where are Raman spectra? In which frequencies and intensities? Please explain it
Thanks for the comment. The results and analysis of Raman spectroscopy are in the Section 3.3.4. The laser wavelength was 532 nm and the wavelength range was 200-2000 nm-1, and we have added the information in the new manuscript, please see Lines 318-319, Page 12.
- Please increase the quality of Figure 3. The parameters in the x and y axes are not clear. What is the total consequence of Figure 3 regarding the similarities anddifferences of the graphs? Please describe it.
Thanks for your suggestion. We have improved the quality of the figure in the revised manuscript. Due to the adjustment of the manuscript structure, please see Fig. 6 on Page 9. Meanwhile, we have added a description of the result of the figure, please see Lines 248-250, Page 9.
- Is titanium (Ti) as a light transition element can be an appropriate option as the additive to aluminum (Al) alloy?
Thanks for the comment. Ti is indeed an effective alloying element in the Al alloy. Ti can be combined with Al to form Al3Ti phases, which can effectively strengthen the aluminum matrix, thereby improving the strength and stiffness of aluminum alloy. The addition of Ti can also form a stable titanium oxide film and enhance the corrosion resistance of the alloy. Meanwhile, Ti can reduce the hot deformation temperature of aluminum alloy and improve the fluidity of the material in the hot-pressing process. Besides, when added to the coatings, the Ti element can effectively release the residual stress by changing the structure of the DLC film.
- Please increase the quality of Figure 4. The parameters in the x and y axes are not clear.
Thanks for your suggestion. We have improved the quality of the original Figure 4. Due to the adjustment of the article structure in the manuscript, the revised figure is shown in Figure 5, please see Lines 231-233, Page 8.
- Page 8, lines 224 & 225: “Combined with the EDS elemental distribution, it is evident that the distribution of O, Al, and P is relatively uniform, and there is no significant difference”. However, the properties of these elements are different. How can the authors illustrate it?
Thanks for the comment. The properties of these elements are indeed different, but the content of each element can be obtained by the location and intensity of different peaks in the EDS spectrum. From the surface distribution of element content, it can be known if there was an enrichment of certain elements in a certain location.
- The experimental conditions including temperature and pressure have not been discussed.
Thanks for the comment. The wear tests were carried out at room temperature and atmospheric pressure. We have added it in the new manuscript, please see Lines 114-115, Page 3.
- The authors have employed a variety of metal and metalloids atoms with different physical and chemical characteristics. Therefore, please mention the limitations of your work.
In this work, the surface hardness and wear resistance of aluminum alloy has been significantly improved by preparing three surface-modified coatings. Compared with the MAO coating, HA+W+DLC, and HA+Ti+ ta-C coatings exhibited better hardness and wear resistance. However, the deposition rates of these two coatings were slow and the binding force was limited. Therefore, these two problems still need to be further optimized in the later work.

Reviewer 2 Report
Comments and Suggestions for Authors
1) The title should be changed. This work is focused on improving the wear properties and not the mechanical properties.
2) What is the novelty of this work? You are not explaining why you selected to use multiple different surface modification processes to improve the wear resistance of Al. Are there any references on similar works? What are the advantages and disadvantages?
3) You provide no information on the surface modification processes you used! How did you perform MAO and HA? What kind of coating did you deposit with PVD? Was it DLC? Ti? TaC? What are the technical details of the deposition?
4) You should provide SEM images of the cross sections of the different coating configurations. Analyse how good the adhesion is, if there are any interactions between the different coatings. Are there any intermediate layers formed?
5) It`s very odd and unusual that you used petroleum to clean the samples after testing. This may have affected the state of the worn surface.
6) It would be beneficial first to present and discuss the wear results and then present the worn surfaces.
7) Unfortunately it is very difficult to assess the results because the information on the surface modification processes are non-existant.
Take all the above into consideration, unfortunately this work has to be rejected.
Author Response
Corrections made for Coatings-2929440
Original Title: Improving the mechanical properties of 7A04 aluminum alloy by three surface modification coatings
Dear editor and reviewer,
We appreciate the helpful comments of the reviewers. The manuscript has been carefully revised in light of these comments, and we hope that the improved manuscript is acceptable for publication. Every change was outlined in the red color in the revised manuscript. The responses to comments are listed as follows (The reviewer’s comments are in italic).
- 1. The title should be changed. This work is focused on improving the wear properties and not the mechanical properties.
Thank you for your suggestion. We have changed the title in the new manuscript, which is “Improving the wear resistance of 7A04 aluminum alloy by three kinds of surface modification coatings”.
- 2. What is the novelty of this work? You are not explaining why you selected to use multiple different surface modification processes to improve the wear resistance of Al. Are there any references on similar works? What are the advantages and disadvantages?
Thank you for the comment. The innovation of the work is not clearly pointed out in the original manuscript, and we have revised the Introduction in the new manuscript. Through literature research (See references 22-27) and pre-research experiments, the three surface modification methods can effectively improve the hardness and wear resistance of aluminum alloy, thereby we chose these three coatings in this work. However, the single method has more or less problems, such as poor surface quality of MAO coating and poor bonding force of DLC coating, the comparison of these three types of coated aluminum alloy is still scarce, and which is the best choice is still worth studying. Therefore, we prepared three coatings using proven techniques in this work, and investigated the microstructure, hardness, and wear behavior of three coatings.
- 3. You provide no information on the surface modification processes you used! How did you perform MAO and HA? What kind of coating did you deposit with PVD? Was it DLC? Ti? TaC? What are the technical details of the deposition?
Thanks for your constructive suggestion. We have supplemented the surface modification processes in Section 2.2 of the new manuscript, please see Lines 98-111, Page 3.
- 4.You should provide SEM images of the cross sections of the different coating configurations. Analyse how good the adhesion is, if there are any interactions between the different coatings. Are there any intermediate layers formed?
Thanks for the nice suggestion. The SEM cross-sectional morphologies of the samples before wear are shown in Fig.2 of the new manuscript. It can be seen that the three coat-ings are well bonded with the substrate, and the interfaces of each sublayer in the two multilayer coatings are clear. HA+W+DLC coating consists of three sublayers of Cr2O3, W, and DLC, and HA+Ti+ ta-C coating consists of three sublayers of NiO, Ti, and ta-C. The metal sublayer (W/Ti) and the carbon layer (DLC/ta-C) were prepared by physical vapor deposition, in which no chemical reactions took place and therefore, no intermediate layer was formed. In addition, we measured the binding forces between the coating and the substrate, as shown in Fig.S1. The binding forces between three coatings and the substrate are 48 N, 16 N, and 20 N, respectively.
Fig.S1 Frictional force and acoustic signal curves (a)MAO coating (b)HA +W+DLC coating (c) HA+Ti+ta-C coating
- 5. It`s very odd and unusual that you used petroleum to clean the samples after testing. This may have affected the state of the worn surface.
Thanks for the comment. Our tests were carried out in the process of oil lubrication. The use of petroleum ether helps to remove the surface lubricating oil and other pollutants. Before observation, the samples were ultrasonic cleaned in anhydrous ethanol for 10 minutes.
- 6.It would be beneficial first to present and discuss the wear results and then present the worn surfaces.
Thanks for your constructive suggestion. We have adjusted the structure of the article according to your nice suggestion. Please see Section 3.3.3 in the new manuscript.
- Unfortunately it is very difficult to assess the results because the information on the surface modification processes are non-existant.
Thanks for the comment. We have supplemented the surface modification processes in Section 2.2 of the new manuscript.

Reviewer 3 Report
Comments and Suggestions for Authors
The article is devoted to the analysis of properties of various coatings for aluminum alloy 7A04 and their influence on hardness and wear resistance. The obtained data have scientific novelty and can be useful.
There are a few minor comments on the content of the manuscript, which should be corrected before the publication process. The main ones are as follows:
1) The introduction should state why aluminum alloy 7A04 was chosen as the base material. It should also state what is the difference between the authors' proposed work and previous studies.
2) Please increase the font size for the captions in Figure 3.
3) In the conclusion, the authors should give recommendations on how to utilize the findings in real industry.
4) Is there any information on how strong a bond is formed between the matrix and the coating? Have evaluative tests of the bond strength been carried out?
5) It is generally accepted that the EMPA method is not suitable for analyzing chemical elements such as oxygen, hydrogen, nitrogen and some others. How were these elements determined in this study?
Comments on the Quality of English LanguageMinor corrections of grammatical errors and typos are required.
Author Response
Corrections made for Coatings-2929440
Original Title: Improving the mechanical properties of 7A04 aluminum alloy by three surface modification coatings
Dear editor and reviewer,
We appreciate the helpful comments of the reviewers. The manuscript has been carefully revised in light of these comments, and we hope that the improved manuscript is acceptable for publication. Every change was outlined in the red color in the revised manuscript. The responses to comments are listed as follows (The reviewer’s comments are in italic).
- 1.The introduction should state why aluminum alloy 7A04 was chosen as the base material. It should also state what is the difference between the authors' proposed work and previous studies.
Thanks for your constructive suggestion. We have added the introduction of 7A04 aluminum alloy in the new manuscript, please see Lines 34-36, Page 1. In addition, the difference between this work and previous studies was introduced in Lines 78-81, Page 2.
- 2.Please increase the font size for the captions in Figure 3.
Thanks for your suggestion. We have improved the quality of Figure 3 in the revised manuscript. Due to the adjustment of the manuscript structure, please see Fig. 6 on Page 9.
- 3.In the conclusion, the authors should give recommendations on how to utilize the findings in real industry.
Thanks for your friendly suggestion. We have rewritten the conclusions in the new manuscript and as follows.
(1) In this work, MAO, HA+W+DLC, and HA+Ti+ta-C coatings were successfully prepared on 7A04 aluminum alloy substrate. The surface hardness was significantly improved after coating. The hardnesses of three coated samples were 7.33 GPa, 22.64 GPa, and 34.23 GPa, respectively. The highest hardness of the HA+Ti+ta-C coatings resulted from the high sp3 C-C bond content.
(2) During the ring-block wear tests under oil lubrication, both two multilayer coatings exhibited excellent wear resistance. The average coefficient of friction and wear rate of HA+W+DLC, and HA+Ti+ta-C were respectively 0.028 and 1.51×10-7 mm3/N·m, 0.025 and 1.36×10-7 mm3/N·m. The higher surface hardness of the HA+Ti+ta-C coating leads to better wear resistance, which makes the coating expected to be applied in the surface protection of aluminum alloys.
- 4.Is there any information on how strong a bond is formed between the matrix and the coating? Have evaluative tests of the bond strength been carried out?
Thanks for the comment. We measured the binding forces between the coating and the substrate, as shown Fig.S1. The binding forces between three coatings and the substrate are 48 N, 16 N, and 20 N, respectively.
Fig.S1 Frictional force and acoustic signal curves (a)MAO coating (b)HA +W+DLC coating (c) HA+Ti+ta-C coating
- 5.It is generally accepted that the EMPA method is not suitable for analyzing chemical elements such as oxygen, hydrogen, nitrogen and some others. How were these elements determined in this study?
Thanks for the comment. EMPA method is indeed not suitable for analyzing the light elements. However, the only concerned light element in the elemental mapping distributions and the line-scanning is oxygen in this work, and we did not need to know the exact content of oxygen, just compared it to the content of other elements.

Reviewer 4 Report
Comments and Suggestions for Authors
The article is devoted to the study of coatings on aluminum alloy grade 7A04. The article has shortcomings and, in my opinion, requires significant improvement. I have several comments that I hope will help the authors improve the article and publish it in the highly rated scientific journal Coatings.
1. The title of the article is formulated poorly. I recommend that you indicate the type of coating being studied in the title. In the study, only hardness was studied among the mechanical properties.
2. The research plan is not clear from 2. Test materials and methods. For what purposes is a ring made of 25Cr3Mo3NiNbZr steel used?
3. The methods do not describe the coating method. The equipment, modes and chemical compositions for applying coatings are not specified. This is very important in this article. I recommend providing this information. How are MAO, HA+W+DLC, and HA+Ti+ta-C coatings applied? What is their difference?
4. How was the surface of the samples prepared before coating?
5. When studying coatings, it is customary to provide photographs from an optical microscope to assess the thickness of the resulting coating, porosity, and the quality of adhesion of the coating to the base metal. This paper does not contain these photos and does not indicate the thickness of the coating.
6. The photos in Fig. 1 do not provide an understanding of the structure of the coating.
7. What is the thickness of the coating? How much is 10% of the thickness? I recommend providing a diagram for measuring hardness on samples.
8. Why were the coatings under study not assessed for corrosion resistance?
9. The reference does not contain articles for the last 2-3 years.
10. The conclusions say “it is shown that the HA+Ti+ta+C coating has higher densities and lower porosity,” but the article does not contain these studies.
11. In my opinion, the conclusions are poorly formulated and do not reflect the essence of the research conducted.
Author Response
Corrections made for Coatings-2929440
Original Title: Improving the mechanical properties of 7A04 aluminum alloy by three surface modification coatings
Dear editor and reviewer,
We appreciate the helpful comments of the reviewers. The manuscript has been carefully revised in light of these comments, and we hope that the improved manuscript is acceptable for publication. Every change was outlined in the red color in the revised manuscript. The responses to comments are listed as follows (The reviewer’s comments are in italic).
- The title of the article is formulated poorly. I recommend that you indicate the type of coating being studied in the title. In the study, only hardness was studied among the mechanical properties.
Thank you for your friendly suggestion. We have modified the title in the new manuscript, which is “Improving the wear resistance of 7A04 aluminum alloy by three kinds of surface modification coatings”.
2.The research plan is not clear from 2. Test materials and methods. For what purposes is a ring made of 25Cr3Mo3NiNbZr steel used?
Thanks for the comment. We have supplemented the research plan of Section 2.2 in the new manuscript, please see Lines 98-111, Page 3. 25Cr3Mo3NiNbZr steel has high strength and toughness, so it can withstand large load and impact loads. It is suitable for the use of wear parts in the wear process.
- The methods do not describe the coating method. The equipment, modes and chemical compositions for applying coatings are not specified. This is very important in this article. I recommend providing this information. How are MAO, HA+W+DLC, and HA+Ti+ta-C coatings applied? What is their difference?
Thanks for your constructive suggestion. We have added a detailed process of coating preparation to the manuscript. Please see Lines 98-111, Page 3.
- How was the surface of the samples prepared before coating?
Thanks for the comment. Prior to coating deposition, the 7A04 aluminum alloy substrates were polished with 2000 mesh sandpaper, which was consistent with the actual working conditions. We have added the description in the new manuscript, please see Lines 98-99, Page 3.
- When studying coatings, it is customary to provide photographs from an optical microscope to assess the thickness of the resulting coating, porosity, and the quality of adhesion of the coating to the base metal. This paper does not contain these photos and does not indicate the thickness of the coating.
Thanks for your nice suggestion. We have added the SEM cross-sectional morphologies of as-deposited coatings in Fig.2 of the new manuscript. The thickness of as-deposited MAO coating, HA+W+DLC coating, and HA+Ti+ta-C coating is 37.1 μm, 9.8 μm, and 3.2 μm, respectively. Three coatings are well bonded with the substrate, and the interfaces of each sublayer in the two multilayer coatings are clear. Please see Lines 163-165, Page 5.
- The photos in Fig. 1 do not provide an understanding of the structure of the coating.
Thank you for your suggestion. We have added the SEM cross-sectional morphologies of as-deposited coatings in Fig.2. Three coatings are well bonded with the substrate, and the interfaces of each sublayer in the two multilayer coatings are clear, please see Lines 163-165, Page 5.
- What is the thickness of the coating? How much is 10% of the thickness? I recommend providing a diagram for measuring hardness on samples.
Thanks for your constructive suggestion. In the new manuscript, we have added the SEM cross-sectional morphologies of as-deposited coatings in Fig.2. The thickness of as-deposited MAO coating, HA+W+DLC coating, and HA+Ti+ta-C coating is 37.1 μm, 9.8 μm and 3.2 μm, respectively. Besides, We have added the sample hardness in the following Table S1.
Table S1 Hardness table of different samples
|
|
Hardness (GPa) |
|||
|
7A04 |
MAO |
HA+W+DLC |
HA+Ti+ta-C |
|
|
1 |
1.74 |
7.17 |
23.46 |
34.33 |
|
2 |
1.75 |
7.71 |
22.25 |
34.47 |
|
3 |
1.73 |
7.01 |
22.21 |
33.89 |
|
4 |
1.76 |
8.25 |
23.92 |
35.10 |
|
5 |
1.70 |
6.58 |
22.17 |
33.26 |
8.Why were the coatings under study not assessed for corrosion resistance?
Thanks for the comment. The purpose of this paper is to compare the wear resistance of the three coatings, and their wear resistance will be shown in another paper.
9.The reference does not contain articles for the last 2-3 years.
Thank you for your friendly suggestion. According to the research content, we have supplemented some literature in the last 2-3 years in the new manuscript, as shown below.
[22] Soffritti C, Fortini A, Nastruzzi A, et al. Dry sliding behavior of an aluminum alloy after innovative hard anodizing treatments[J]. Materials, 2021, 14(12): 3281.
[23] Kwolek P, ObÅ‚ój A, KoÅ›cielniak B, et al. Wear resistance of hard anodic coatings fabricated on 5005 and 6061 aluminum alloys[J]. Archives of Civil and Mechanical Engineering, 2024, 24(2): 1-16.
[29] Shen Y, Luo J, Liao B, et al. Tribocorrosion and tribological behavior of Ti-DLC coatings deposited by filtered cathodic vac-uum arc[J]. Diamond and Related Materials, 2022, 125: 108985.
[30] Yi M, Wang T, Liu Z, et al. Tribological Performance of Steel/W-DLC and W-DLC/W-DLC in a Solid–Liquid Lubrication System Additivated with Ultrathin MoS2 Nanosheets[J]. Lubricants, 2023, 11(10): 433.
[42] Cao H, Ye X, Li H, et al. Microstructure, mechanical and tribological properties of multilayer Ti-DLC thick films on Al alloys by filtered cathodic vacuum arc technology[J]. Materials & Design, 2021, 198: 109320.
- The conclusions say “it is shown that the HA+Ti+ta+C coating has higher densities and lower porosity,” but the article does not contain these studies.
Thanks for the comment. We are sorry for the flawed conclusions in the original manuscript, and we have rewritten it in the new manuscript.
- In my opinion, the conclusions are poorly formulated and do not reflect the essence of the research conducted.
Thanks for the comment. We have rewritten the conclusions in the new manuscript and as follows.
(1) In this work, MAO, HA+W+DLC, and HA+Ti+ta-C coatings were successfully prepared on 7A04 aluminum alloy substrate. The surface hardness was significantly improved after coating. The hardnesses of three coated samples were 7.33 GPa, 22.64 GPa, and 34.23 GPa, respectively. The highest hardness of the HA+Ti+ta-C coatings resulted from the high sp3 C-C bond content.
(2) During the ring-block wear tests under oil lubrication, both two multilayer coatings exhibited excellent wear resistance. The average coefficient of friction and wear rate of HA+W+DLC, and HA+Ti+ta-C were respectively 0.028 and 1.51×10-7 mm3/N·m, 0.025 and 1.36×10-7 mm3/N·m. The higher surface hardness of the HA+Ti+ta-C coating leads to better wear resistance, which makes the coating expected to be applied in the surface protection of aluminum alloys.

Round 2
Reviewer 1 Report
Comments and Suggestions for Authors
Dear Editor,
Regarding the author’s revision, I am pleased to inform my satisfaction of present form of the manuscript entitled: “Improving the mechanical properties of 7A04 aluminum alloy by three surface modification coatings” for publication in Coatings.
Author Response
Dear editors and reviewers,
We are very grateful for your valuable comments and good suggestions, which have greatly helped improve our manuscripts, and for your recognition of our work. We have re-optimized some of the expressions in the article so that readers can read it better.
Reviewer 2 Report
Comments and Suggestions for Authors
The revised manuscript, while the authors made efforts for improvement, still faces the same issues highlighted during the initial review:
1) The authors do not explain why they selected to perform multiple surface modification processes that are very different in nature. Combining different surface modification techniques without a clear scope does not make a work novel.
2) There is no reference on the potential applications of the modified Al alloy.
3) While the authors made effort to expand the experimental section on surface modification processes the section is still confusing. They need to explain every process, step by step in a clear and consistent way.
4) Authors added SEM images of the cross sections but the description is very poor. What is the chemical homogeneity of each modification? Are there any diffusion layers? How porous are those layers? Are they oxidized? Those questions need to be answered to assess the quality of the surface modification.
Taken all the above into consideration unfortunately this work cannot be accepted.
Author Response
Corrections made for Coatings-2929440
Original Title: Improving the mechanical properties of 7A04 aluminum alloy by three surface modification coatings
Dear editors and reviewers,
We appreciate the helpful comments of the reviewers. The manuscript has been carefully revised again in light of these comments, and we hope that the improved manuscript is acceptable for publication. Every change was outlined in the blue color in the revised manuscript of round 2. The responses to comments are listed as follows. (The reviewer’s comments are in italics).
1.The authors do not explain why they selected to perform multiple surface modification processes that are very different in nature. Combining different surface modification techniques without a clear scope does not make a work novel.
Thanks for the comment. The key point of this work is to improve the wear resistance of 7A04 aluminum alloy. Micro-arc oxide coating, DLC coating, and ta-C coating are three commonly used surface-modified coatings to achieve this goal, among which DLC/ta-C needs to add transition layers due to poor binding force. However, the direct comparison of these three kinds of coatings is still scarce. Therefore, it is novel and worth studying the comparison of wear resistance of the three coatings on 7A04 aluminum alloy. This work provides an experimental basis for finding the best choice to improve the wear resistance of 7A04 aluminum alloy.
2.There is no reference on the potential applications of the modified Al alloy.
Thanks for the comment. 7A04 is a kind of superhard aluminum alloy, the dosage of which is second only to steel in modern industry, and it is one of the most widely used aluminum alloys in the aerospace field. Thus, the modified Al alloy is expected to be applied in aerospace materials, such as missiles and high-speed aircraft, which has been depicted in the first paragraph of the Introduction.
3.While the authors made effort to expand the experimental section on surface modification processes the section is still confusing. They need to explain every process, step by step in a clear and consistent way.
Thanks for your nice suggestion. We have revised the experimental section on surface modification processes step by step in the new manuscript, please see Lines 98-114, Page 3.
- Authors added SEM images of the cross sections but the description is very poor. What is the chemical homogeneity of each modification? Are there any diffusion layers? How porous are those layers? Are they oxidized? Those questions need to be answered to assess the quality of the surface modification.
Thanks for the comment. We have revised Fig.2 (as shown below) and the corresponding description in the new manuscript, please see Lines 156-163, Page 4. Three coatings are dense with no pores and well bonded with the substrate, and the interfaces of each sublayer in the two multilayer coatings are clear. There are no distinct diffusion layers. Except for the MAO and HA sublayers in two multilayer coatings, other sublayers are not oxidized. Additionally, the distribution of Al and O elements is relatively uniform in MAO, and the composition of HA+W+DLC and HA+Ti+ta-C conforms to the design, which is conducive to binding with the substrate.
Figure 2. Cross-section morphologies and the corresponding elemental line scans of (a) MAO coating (b) HA+W+DLC coating (c) HA+Ti+ta-C coating

Reviewer 4 Report
Comments and Suggestions for Authors
The authors made significant corrections to the article. I recommend accepting the article for publication in this version.
Author Response

(The authors gave the same response as above.)
